# Ultrasound-Guided Centrally Inserted Central Catheter (CICC) Placement in Newborns: A Safe Clinical Training Program in a Neonatal Intensive Care Unit

**DOI:** 10.3390/children11040395

**Published:** 2024-03-26

**Authors:** Tommaso Zini, Lucia Corso, Cinzia Mazzi, Cecilia Baraldi, Elisa Nieddu, Laura Rinaldi, Francesca Miselli, Luca Bedetti, Eugenio Spaggiari, Katia Rossi, Alberto Berardi, Licia Lugli

**Affiliations:** 1Neonatal Intensive Care Unit, Department of Medical and Surgical Sciences of Mothers, Children and Adults, University of Modena and Reggio Emilia, 41125 Modena, Italy; tommaso.zini@unimore.it (T.Z.); mazzi.cinzia@aou.mo.it (C.M.); baraldi.cecilia@aou.mo.it (C.B.); nieddu.elisa@aou.mo.it (E.N.); 79638@studenti.unimore.it (F.M.); bedetti.luca@aou.mo.it (L.B.); spaggiari.eugenio@aou.mo.it (E.S.); rossi.katia@aou.mo.it (K.R.); lugli.licia@aou.mo.it (L.L.); 2Post-Graduate School of Paediatrics, Department of Medical and Surgical Sciences of Mothers, Children and Adults, University of Modena and Reggio Emilia, 41125 Modena, Italy; 196397@studenti.unimore.it; 3Anaesthesia and Intensive Care Medicine, Department of Medical and Surgical Sciences of Mothers, Children and Adults, University of Modena and Reggio Emilia, 41125 Modena, Italy; rinaldi.laura@aou.mo.it

**Keywords:** central venous catheterization, neonates, critically ill neonates, training, simulation, ultrasound guidance, neonatal intensive care, neonatology

## Abstract

Background: Centrally inserted central catheters (CICCs) are increasingly used in neonatal care. CICCs have garnered attention and adoption owing to their advantageous features. Therefore, achieving clinical competence in ultrasound-guided CICC insertion in term and preterm infants is of paramount importance for neonatologists. A safe clinical training program should include theoretical teaching and clinical practice, simulation and supervised CICC insertions. Methods: We planned a training program for neonatologists for ultrasound-guided CICCs placement at our level III neonatal intensive care unit (NICU) in Modena, Italy. In this single-centre prospective observational study, we present the preliminary results of a 12-month training period. Two paediatric anaesthesiologists participated as trainers, and a multidisciplinary team was established for continuing education, consisting of neonatologists, nurses, and anaesthesiologists. We detail the features of our training program and present the modalities of CICC placement in newborns. Results: The success rate of procedures was 100%. In 80.5% of cases, the insertion was obtained at the first ultrasound-guided venipuncture. No procedure-related complications occurred in neonates (median gestational age 36 weeks, IQR 26–40; median birth weight 1200 g, IQR 622–2930). Three of the six neonatologists (50%) who participated in the clinical training program have achieved good clinical competence. One of them has acquired the necessary skills to in turn supervise other colleagues. Conclusions: Our ongoing clinical training program was safe and effective. Conducting the program within the NICU contributes to the implementation of medical and nursing skills of the entire staff.

## 1. Introduction

### 1.1. Central Venous Catheters (CVCs) in Neonatal Intensive Care Units (NICUs)

Central Venous Catheters (CVCs) are fundamental intravascular devices in preterm and term infants admitted to neonatal intensive care units (NICUs). CVCs are defined as Central Venous Access Devices (CVADs) placed by cannulation of deep veins that lie deeper than 7 mm from the surface of the skin [1]. A CVAD is defined as when the tip is located at the atriocaval junction or in the inferior vena cava [2]. CVCs include Peripherally Inserted Central Catheters (PICCs, placed in deep veins of the arm), Centrally Inserted Central Catheters (CICCs, placed in deep veins of the supra/infraclavicular area), and Femorally Inserted Central Catheters (FICCs, placed in deep veins of the groin). PICCs are different from percutaneously inserted Epicutaneo-Caval Catheters (ECCs), which are CVADs placed in peripheral and superficial veins [3]. Umbilical Venous Catheters (UVCs) are short-term CVADs that can be inserted into the umbilical vein at birth or immediately after birth (within 24 h of life) [4]. Two of the most commonly used CVADs in neonates, ECCs and UVCs, should not properly be considered PICCs or CVCs because they are placed in superficial veins (ECC) and at the umbilical stump (UVC), respectively.

In neonatal intensive care, ECCs are traditionally the most commonly used catheters for administering (i) drugs not suitable via Peripheral Venous Catheters or Short Peripheral Cannulas (PVCs or SPCs) and (ii) for mid-term parenteral nutrition, especially in preterm infants [5]. Therefore, neonatologists are trained and experienced in ECC insertion, as the first CVAD or as a replacement for a previously placed UVC. In addition, neonatologists’ expertise in point-of-care ultrasound (POCUS) applications is growing [6]. POCUS techniques include ultrasound (US) guidance for intra-procedural real-time catheter’s tip navigation and tip location during ECC, UVC, and CVC placement and post-procedural early recognition of secondary malposition due to tip migration [7]. To assess ECC, UVC, and CVC tips, standard chest and abdomen radiographs (X-ray) have some relevant limitations (e.g., relatively inaccurate and post-procedural methodology), whereas real-time US in experienced hands, using structured protocols such as the “Neo-ECHOTIP” protocol, have several potential advantages (e.g., accurate and intra-procedural methodology appropriate for both navigation and tip location of all CVADs used in NICU) [4]. Moreover, the choice of ultrasound has obvious advantages in neonates: it is noninvasive and safe, and the risk of exposure to ionizing radiation is reduced [8,9].

### 1.2. The Choice of the Most Appropriate Venous Access Device (VAD) in Neonates

In recent years, CICCs have gained attention and adoption due to some advantageous characteristics, including that they are power-injectable (e.g., resistant to high pressures and thus able to tolerate high flows and high injection pressures) and suitable for sampling, and they allow transfusion of blood products and hemodynamic monitoring. In Italy, GAVECELT is a leading expert group on venous accesses, and GAVEPED is its paediatric interest group [10]. The GAVECELT panel has published recommendations for choosing the most appropriate Venous Access Device (VAD, or DAV) in potentially complex situations [11]. The “DAV-EXP” algorithm, developed by GAVECELT, is a simple and clear tool to facilitate clinical decision making on a case-by-case basis: http://davexpert.gavecelt.it/ (accessed on 8 February 2024) [10]. It refers to Central VADs (CVADs) indicated for (i) infusion therapies, (ii) hemodynamic monitoring, and/or (iii) frequent and repeated blood samples, while devices placed for blood exchange manoeuvres (e.g., haemodialysis, apheresis, ultrafiltration) are excluded. It covers all age groups, both intensive and non-intensive settings, including neonatal intensive care. GAVECELT experts suggest solutions based on appropriateness in terms of safety for the patients, clinical effectiveness of the CVADs, and cost-effectiveness (appropriate resource allocation).

The part of the algorithm called “Neonatal DAV-Expert” by the GAVECELT/GAVEPED panel is fully focused on newborns. It deals with neonates who require venous access at birth and >24 h after birth separately. At birth, the decision regarding UVC placement is determined by considerations of gestational age (particularly for extremely preterm births) and clinical severity, which may include factors such as severe asphyxia, hemodynamic instability, or the necessity for invasive ventilation (or non-invasive ventilation with “high” parameters). Alternatively, UVC placement may be chosen at birth due to the difficulty in positioning a peripheral cannula (PVC or SPC) [12]. Indications for Umbilical Arterial Catheter (UAC) placement at birth are beyond the scope of this article. After birth, when the insertion of a UVC is no longer possible, the choice of the most appropriate type of VAD depends mainly on (i) the clinical conditions, and (ii) the expected duration of the treatment. In critically ill infants and when more than 14 days of treatment is expected, a CICC (or FICC) insertion under US guidance is usually preferred [13]. In summary, the GAVECELT/GAVEPED consensus listed the appropriate indications for the use of US-guided CICCs (or FICCs), which include common situations such as stable preterm infants with an expected duration of parenteral nutrition longer than 14 days [13]. In addition, the insertion of a CICC (or a FICC) may be indicated in infants with superficial vein depletion, as evidenced by the application of a “RASUVA” (Rapid Superficial Vein Assessment) protocol, or when the insertion of an ECC is difficult or impossible to achieve [14]. As a matter of fact, in order to make a rational choice of the best insertion site (tailored on the single infant and optimized for the specific type of VAD), the GAVECELT/GAVEPED experts recommend a pre-procedural evaluation of all superficial veins of the newborn, with systematic exploration of seven skin areas: medial malleolus, lateral malleolus, retro-popliteal fossa, back of the hand and wrist, antecubital fossa, anterior scalp surface, and posterior scalp surface [14].

Considering all of the aspects mentioned above, there is an increasing number of conditions in which (given the same needs of the individual patient and the resources of the local NICU) the choice of CICCs is deemed advantageous. However, in many settings, CVADs are implanted by consultants only (e.g., paediatric anaesthesiologists), and the standard minimal requirements for training neonatologists are mostly undefined [15]. The acquisition of knowledge and skills in neonatal intensive care requires specific education and training to ensure competent practice and reduce mechanical and infectious complications.

### 1.3. Complications of Central Venous Access Devices (CVADs) in Neonates

Neonatologists are increasingly concerned about complications of CVCs, particularly UVCs, and ECCs. CVC- and ECC-associated complications include primary and secondary dislocation or malposition, infection, thrombosis, emboli, arrhythmia, and organ injury, particularly liver injury [16]. In multinational multicentre surveys, catheter-related complication rates vary among centres and by the type of VAD [17]. However, all experts agree on the need to reduce the risks of infectious and thrombotic complications related to the unnecessary use of UVCs. Therefore, the UVC should be promptly removed when it is no longer needed, and it is reasonable to limit the UVC dwelling time to a maximum of 7 days [18]. In case a CVC is still required, the UVC should be preferably removed early, within 4–5 days, and a new central line should be inserted [19]. Indirectly, the choice of an early UVC removal approach results in an increasing number of indications for US-guided CICC (or FICC) placement.

### 1.4. Training Programs on Central Venous Catheters (CVCs) in Neonatal Care

The World Congress on the Vascular Accesses (WOCOVA) foundation, the global network of associations on vascular accesses, established an international task force to provide an evidence-based consensus on training, insertion, and maintenance of CVADs [6]. A fully standardized program for trainees should be based on both theoretical teaching and clinical practice of insertion procedures, infection prevention, complications, care, and device maintenance. A well-established program should include standardized education, simulations with US, and supervised insertions [20]. Standard didactic sessions should cover basic knowledge of anatomy and physiology, US-guided technique and CVC tip location, infection control strategies and sterile techniques, and catheter selection. Educational courses should focus on both insertion procedures and management of CVADs. Simulation training is the key to safe patient insertions, and educational processes (that include the application of US-guidance) enhance success and safety [21]. Supervised placements are required to establish credentialing for CVAD procedures. The task force listed a total of 16 recommendations for minimal education and training for CVAD insertion and management, with specific recommendations for children and neonates [6]. A validation of competencies requires supervision of a specified number of successful procedures and must be performed using global rating scales and checklists [22]. Assuming that neonatologists working in the NICUs have sufficient experience with CVCs placement in neonates, particularly ECCs, approximately 15 cannulations would be required to feel comfortable with the US-guided supraclavicular approach to the brachiocephalic vein (BCV) [23].

### 1.5. Aim of the Study

At our NICU, since 2017, we have planned a clinical training program for neonatologists for the US-guided placement of CICCs in the BCV. We were inspired by the best practices of the GAVECELT/GAVEPED panel, and our training modalities were based on the recommendations of the international evidence-based consensus task force (WOCOVA) [6,24]. The purpose of our study is to analyse the safety and effectiveness of our training program. Our goal is to implement our clinical training program and provide a useful model for other NICUs.

## 2. Materials and Methods

### 2.1. Study Design

This is a single-centre prospective observational study. The clinical training program was carried out at the Neonatal Intensive Care Unit (NICU) of the University Hospital of Modena, Italy. Our NICU is a tertiary care centre with approximately 400 admissions per year, nearly 40 of whom are very-low-birth-weight (VLBW) infants. Our NICU staff consists of 12 neonatologists and 30 nurses. This study investigates the training process of 6 of our neonatologists in learning how to properly perform US-guided CICC placement. The trainers were a team of experts in the field, as detailed below. The preliminary data presented in this paper cover a 12-month period of the “on-the-job training phase”, from 1 January to 31 December 2022. Approval was obtained from the local Ethics Committee of the Vast Area of North Emilia (Protocol 1340/2020/OSS*/AOUMO). Written informed consent was obtained from parents of enrolled patients.

### 2.2. Training Program

A multidisciplinary team on CVCs, consisting of 2 neonatologists, 4 nurses, and 2 paediatric anaesthesiologists as trainers, was established. The “CVC team” was responsible for creating and periodically updating instructions for CVC insertion procedures and CVAD maintenance. Moreover, the “CVC team” monitored the progress of the clinical training program for neonatologists. During the study period, 6 neonatologists were selected to receive training on US-guided CICC placement. The 6 neonatologists in training were already experienced in bedside ultrasound: 3 were experts in neonatal functional echocardiography, 2 in cranial ultrasound, and 1 in lung ultrasound. Throughout the procedures, the neonatologists and paediatric anaesthesiologists received continuous active assistance from all the NICU nurses. As a matter of fact, all of the nurses had been trained to assist doctors in the placement of CVCs. One of the nurses, who is part of the “CVC team”, has also attended courses to perform CVC placement as a primary operator.

The clinical training program was consistent with the guidelines outlined in the WOCOVA consensus [6] and modulated on the experience of our paediatric anaesthetists. The steps followed in our training program were organized as follows:Training began with a theoretical course (GAVECELT), conducted between 2017 and 2018 (https://gavecelt.it).Theoretical and practical courses, including simulations of US-guided CICC insertion in infants and children, were conducted in 2019, with the participation of 2 experienced paediatric anaesthesiologists as trainers and the entire CVC team.At the same time, from 2019 to 2021, bedside US-guided CICC placements in newborns were performed in our NICU by 2 experienced paediatric anaesthesiologists as reference trainers.In 2022, the 6 neonatologists with expertise in bedside ultrasound, who had been selected, participated in the CICC placements as observers alongside paediatric anaesthesiologists (“observer-neonatologists”—Level of Autonomy 1).After neonatologists observed at least 5 procedures on newborns in the NICU, they began to perform US-guided CICC insertions under the direct supervision of paediatric anaesthesiologists (“in-training-neonatologists”—Level of Autonomy 2).After neonatologists performed at least 3 correct CICC insertions (under direct supervision) and were deemed suitable by the paediatric anaesthesiologists (based on a checklist of acquired knowledge and skills), they could perform US-guided CICC insertions autonomously (“trained-neonatologists”—Level of Autonomy 3).After at least 10 successful US-guided CICC placements, the neonatologists reached a higher level of autonomy and were able to assist further colleagues in their training (“experienced-neonatologists”—Level of Autonomy 4).

### 2.3. CICC Placement Procedure

The CICC placement procedures (Figure 1 and Figure 2) were performed under sedo-analgesia with ketamine (1–2 mg/kg) and midazolam (50–150 μg/kg) for infants ≥33 weeks gestational age (GA) or fentanyl (1–2 μg/kg) for infants <33 weeks GA. In addition, a pacifier with 24% sucrose per os (0.2–0.5 mL or 1–2 mL for preterm and term infants, respectively) was offered in the preparatory phase, a quiet environment was provided (noise was reduced), and appropriate position and containment were achieved. Proper local anaesthesia at the CVC insertion site in case of tunnelling, with subcutaneous infiltration of lidocaine 1% (buffered lidocaine solution with sodium bicarbonate; maximum dose 0.5 mL/kg), has also been used.

During the procedures, the neonate was positioned supine, with the neck slightly extended and the head turned towards the opposite side of the chosen BCV (Figure 1A). The primary operator was positioned on the same side as the venipuncture site (Figure 1A). Intraprocedural ultrasound was performed with a Philips’ L15-7io linear array intraoperative hockey stick US probe transducer connected to a Philips EPIQ 7C ultrasound unit. Power-injectable, polyurethane non-valved catheters of 22 G—2 Fr single lumen or 3 Fr double lumen were placed (60 mm length). The optimal catheter size was selected following the indications by the WOCOVA-GAVECELT-WINFOCUS consensus and the INS standards, which recommend a catheter to vein ratio of 1:3 [25,26,27]. We measured the vein’s diameter in order to match it with the catheter size and reduce the risk of venous thrombosis (Figure 1B,C).

The “RACEVA” (Rapid Central Vein Assessment) protocol was employed for the systematic ultrasound examination of the veins in the supra-clavicular/sub-clavicular area prior to CICC placements (Figure 1) [28]. Following this, skin antisepsis with single-dose 2% chlorhexidine (2% chlorhexidine gluconate in 70% isopropyl alcohol solution) was performed, and a sterile field was established (Figure 2(A.2)). Subsequently, US-guided identification of the BCV (with sterile probe cover) and its cannulation via the supraclavicular approach was conducted (Figure 2B), using the modified Seldinger technique (Figure 2(C.1)) with micro-introducer dilators of appropriate calibre (Figure 2(C.2)). The direction of the guidewire into the vasculature was assessed via US (Figure 2D, Appendix A), followed by tip localization using echocardiography. High-viscosity, 2-octyl cyanoacrylate glue (0.36–1.0 mL), which provides an immediate haemostatic effect, was then utilized to close the puncture site and seal the exit site (Figure 2E). Finally, the catheter was secured with a suture-less dressing and securement device, and the exit site was covered with a semipermeable transparent membrane (Figure 2F).

Before the catheter was used for fluid and medication infusion, the correct tip placement at the atriocaval junction was verified through echocardiography. Procedures were defined as successful if they resulted in correct CICC placement, regardless of the number of US-guided venipunctures performed.

An insertion bundle checklist has been prepared for operators to complete for each CICC placement (Table 1).

## 3. Results

The clinical training program is still ongoing. Over a 12-month period, the six neonatologists achieved different autonomy levels in the bedside US-guided CICC placement on preterm and term infants (Levels of Autonomy 2–4). Table 2 shows the results obtained with a total of 41 procedures performed under the training program.

The age of neonatologists ranged from 37 to 56 years (min–max), and the years of experience in neonatal care ranged from 5 to 24 years (min–max). In 12 months, three of the six neonatologists (50%) who participated in the clinical training program achieved good clinical competence and are now considered “trained-neonatologists” (n = 2, Level of Autonomy 3) or “experienced-neonatologist” (n = 1, Level of Autonomy 4), and the last one is now mentoring colleagues. Although not specifically included in the training program, one neonatologist has also acquired the competence in subcutaneous catheter tunnelling.

This training program has yielded excellent results. In fact, the success rate of the procedure was 100%. Moreover, in 80.5% of cases, CICC placement was successful at the first US-guided venipuncture. The insertion bundle checklist was fulfilled in all cases. No procedure-related complications occurred. Successful rates did not differ in the subgroup of extremely-to-very-preterm (<32 weeks postmenstrual age) and very-low-birth-weight (<1500 g birth weight) infants (n = 15), compared with infants of higher gestational age and birth weight (n = 26).

We distinguished the newborns for whom US-guided CICC placement was indicated based on the categories of neonatal disorders (i.e., based on the discharge diagnosis). We listed the following diagnostic categories (in decreasing order of prevalence in our population): extreme prematurity (n = 15), surgical conditions (n = 11), genetic disorders and congenital malformations (n = 6), infections (n = 5), respiratory disorders (n = 3), and hypoxic-ischemic encephalopathy (n = 1). Extreme prematurity accounted for more than a third of the total (36.6%); the first three categories together (extreme prematurity, surgical conditions, genetic disorders and congenital malformations) represented over three-quarters of the total (78.0%).

## 4. Discussion

We presented preliminary results from the first 12 months of a safe clinical training program for CICC placement carried out in our third level NICU at the University Hospital of Modena.

Training in CICC placement in newborns typically involves a combination of theoretical knowledge and practical skills. In our training program, neonatologists underwent specialized training to ensure safe and effective CICC insertions in infants. The placement of CICCs in neonates is particularly challenging compared to other life stages, due to the small size of the patient and, consequently, the anatomical structures. Given the crucial importance of understanding the anatomy and physiology of preterm and full-term newborns, a neonatologist can be considered a suitable candidate for training, as they possess specialized knowledge in this area. Furthermore, the neonatologist is well versed in the medical conditions that may necessitate CICC placement and is acquainted with situations where it might be contraindicated. A thorough understanding of the vascular system, particularly identifying suitable veins for CVC placement, and being aware of potential complications and risks linked to catheter insertion are essential. This encompasses considerations such as infection, thrombosis, and other complications specific to newborns. Moreover, neonatologists are familiar with established guidelines and protocols for CVC placement and management in newborns. These guidelines help to ensure standardized and safe practices.

In our program, we included simulation-based training, allowing neonatologists to practice CICC placement. This helped them to develop and refine their skills in a controlled environment. Trainees gained practical experience under the supervision of experienced practitioners. This hands-on practice occurred in the NICUs and was crucial for developing proficiency and succeeding in the procedure. In-place practical supervision became feasible thanks to interdisciplinary collaboration. The training included collaboration with paediatric anaesthesiologists who are experts in CICCs placement and are involved in the care of newborns requiring CICCs. Thanks to the supervision of two “consultant trainers”, three neonatologists of our staff have become proficient in placing CICCs independently. One of them has acquired the necessary skills to in turn supervise other colleagues (“experienced-neonatologist”). We believe that this disparity in the achievement of training objectives (i.e., different levels of autonomy were achieved, from 2 to 4) is difficult to avoid in real life. In fact, it is necessary to match the proper timing of procedures (urgent and programmable ones), the priority needs of newborns, and the needs of health professionals (including nurses, “in-training-neonatologists”, “consultant trainers”, i.e., the two paediatric anaesthesiologists). Nevertheless, our training can be envisioned as a hierarchically structured and sustainable program in which those who become proficient in placement can, in turn, train and mentor other colleagues. Therefore, the remaining three “in-training-neonatologists” will more easily complete their training program by the end of 2024, and the rest of the staff can undertake the same program supported by both paediatric anaesthesiologists and trained colleagues.

In our training program, no complications related to catheter placement were observed, the success rate of the procedure was 100%, and 80.5% catheters were successfully inserted at the very first venipuncture. This success rate is close to the performance previously reported by GAVECELT/GAVEPED experts, in which only two out of thirty VLBW infants (6.7%) required a second attempt performed on the same vein [29,30]. This can be considered evidence of the effectiveness and safety of our training program. In addition, the insertion bundle checklist was followed and completed in all cases, attesting to the adoption and systematic application of evidence-based practices in CVC insertion.

Given the small sample size, we cannot draw conclusions about the effectiveness of the program based on the characteristics of the trainees (age of neonatologists, or years of experience in neonatal care). Similarly, comparisons between subgroups of newborns are not significant. However, we observed that there was no obvious neonatal condition (gestational age, birth weight, days of life, diagnostic category) that was clearly associated with a drastically increased risk of procedure failure and/or complications. Interestingly, the categories of neonatal disorders in which indications for CICC placement occur are, in most cases, a reason for long-term hospitalization in the NICU (extreme prematurity, surgical conditions, genetic disorders, and congenital malformations). In our experience, applying the “Neonatal DAV-Expert” algorithm after 24 h of life, the main reason why CICC is chosen as the most appropriate type of VAD is when a long duration of treatment is expected (as typical of conditions above).

Limitations of our program are that neither the catheter tunnelling nor intracavitary ECG methods for the evaluation of correct tip location were specifically included in the training. Nevertheless, during the study period, one neonatologist has also acquired the competence in subcutaneous catheter tunnelling. Furthermore, these skills will be addressed in subsequent training sessions. Given the evolving nature of medical practices, ongoing education and staying updated on the latest advancements are crucial. Additionally, adherence to ethical considerations, patient safety, and a commitment to evidence-based practices are integral components of CICC placement training in newborns.

## 5. Conclusions

CICCs are of increasing use and importance in the NICUs. Achieving clinical competence in ultrasound-guided CICC insertion in term and preterm infants is of paramount importance. Our clinical training program for neonatologists proved to be safe and effective. Our training program included standardized theoretical teaching and clinical practice, simulation practices and supervised CICC insertions. Conducting the training program within the NICU contributes to the implementation of multidisciplinary medical-nursing skills of the entire staff for CVC insertion procedures and device maintenance.

## Figures and Tables

**Figure 1 children-11-00395-f001:**
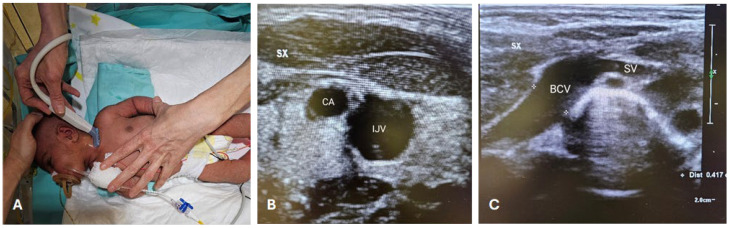
RACEVA protocol. Pre-operative US scan with the so-called RACEVA (Rapid Central Vein Assessment) protocol. (**A**) Systematic US examination of the veins in the supra-clavicular/sub-clavicular area; (**B**) US localization of CA and IJV; (**C**) US localization of BCV and SV. Abbreviations. BCV: brachio-cephalic vein; CA: carotid artery; IJV: internal jugular vein; SV: subclavian vein; SX: left; US: ultrasound.

**Figure 2 children-11-00395-f002:**
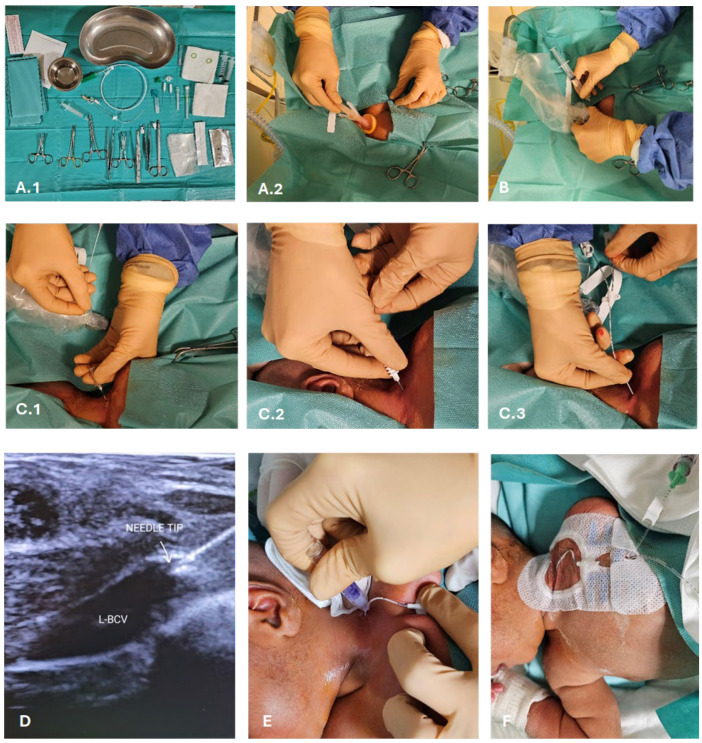
CICC placement procedure. (**A**) Sterile field preparation (**A.1**), barrier precautions and skin antisepsis with 2% chlorhexidine in 70% isopropyl alcohol (**A.2**); (**B**) US-guided venipuncture of the BCV by the supraclavicular approach (visualization in the long-axis); (**C**) vein cannulation using the modified Seldinger technique (**C.1**) and a micro-introducer kit (**C.2**), introduction of polyurethane 3 Fr double lumen catheter (**C.3**); (**D**) US-based tip navigation; (**E**) application of cyanoacrylate glue for the closure of the puncture site and for the sealing of the exit site; (**F**) securement with suture-less device and coverage of the exit site with a transparent membrane. Abbreviations: L-BCV (left-brachiocephalic vein).

**Table 1 children-11-00395-t001:** CICC insertion bundle checklist.

CICC Placement Procedure	Check (“X”) If the Item Has Been Executed
Obtaining written informed consent from parents prior to implantation of CICC	
Preliminary ultrasound assessment of the patient’s venous anatomy using standardized protocols (RACEVA) to choose the vein	
Adequate pharmacological sedation and analgesia of the newborn	
Maximum barrier precautions, including face mask, headcover, hand hygiene, sterile gown, and sterile gloves	
Disinfection with single-dose 2% chlorhexidine and sterile field preparation	
Ultrasound-guided venipuncture to visualize the needle progression from the skin surface to the target vein, avoiding accidental damage to surrounding structures	
Intraprocedural localization of the catheter tip by transthoracic echocardiography to ensure correct tip placement during implantation	
Catheter stabilization by application of cyanoacrylate glue	
Application of a transparent, semipermeable membrane	

**Table 2 children-11-00395-t002:** Results. Demographics and results of ultrasound-guided CICC placements in the clinical training program at the NICU of Modena, 2022.

Neonatologists	
Median age, years (IQR)	47 (41–54)
Median experience in neonatal care, years (IQR)	21 (11–22)
**Newborns**	
Median gestational age, weeks (IQR)	36 (26–40)
Median birth weight, g (IQR)	1200 (622–2930)
Median days of life at insertion, days (IQR)	16 (5–33)
**US-guided CICC placements**	
Number of procedures in the training program, n	41
Successful procedure, n (%)	41 (100)
Successful insertion at first venipuncture, n (%)	33 (80.5)
Procedure-related complications, n (%)	0 (0)
Dislocation or malposition, n	0
Infection, n	0
Thrombosis, n	0
Emboli, n	0
Arrhythmia, n	0
Organ injury, n	0
Elective removal, n (%)	41 (100)

Abbreviations. CICC: centrally inserted central catheter; IQR: interquartile range; NICU: neonatal intensive care unit; US: ultrasound.

## Data Availability

De-identified individual participant data presented in this study are available on request from the corresponding author. The data are not publicly available due to the need for use in further research.

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
