# Peer review of "Ultrasound-Guided Centrally Inserted Central Catheter (CICC) Placement in Newborns: A Safe Clinical Training Program in a Neonatal Intensive Care Unit"

_children, 2024, doi:10.3390/children11040395_

Round 1

Reviewer 1 Report

Comments and Suggestions for Authors

I assume that the majority of neonatologists and neonatal units is aware of the increased feasibility and safety of ultrasound guided deep venous access acquisition in (pre)term neonates, so that neonatologists also have to acquire the skills needed. The authors describe their training and educational program for a specific number of cases.

There is value in this paper, but this is likely ‘only one’ of the potential approaches. Consequently, we do need much more details on the approach taken, and this reviewer would encourage the authors to e.g. add supplements to better understand the educational part of the program, and to add more details on materials, equipments and drugs (doses?) applied. If there are videos available, there is very likely add on benefit to add these.

Furthermore, it is a missed opportunity to only report on your expertise until end 2022, and another year could be added, to bring more ‘sustained’ information on the teaching implementation. This is even more the case as you mention that the clinical training is still ongoing.

How has ‘successful procedure’ been described ?

Specific

Patient consent (or parental) reads somewhat bizar and contradicting between the additional info to the paper.

The same holds true for the data availability statement, I cannot see the relevance of the second sentence (as also publicly available data can be use of additional research).

Author Response

Thank you very much for taking the time to review this manuscript.

We also thank you for your comments and suggestions, which we have taken into account to improve the quality of the manuscript.

Please find the detailed responses below and the corresponding revisions in the re-submitted file.

There is value in this paper, but this is likely ‘only one’ of the potential approaches. Consequently, we do need much more details on the approach taken, and this reviewer would encourage the authors to e.g. add supplements to better understand the educational part of the program, and to add more details on materials, equipments and drugs (doses?) applied. If there are videos available, there is very likely add on benefit to add these.

  1. We improved the Materials and Methods section. We added Supplementary Materials (Video S1: US-guided needle insertion; Video S2: US-guided needle insertion training). We hope that the resubmitted manuscript is more appreciable than what was commented on and suggested by the reviewer.

Furthermore, it is a missed opportunity to only report on your expertise until end 2022, and another year could be added, to bring more ‘sustained’ information on the teaching implementation. This is even more the case as you mention that the clinical training is still ongoing.

  1. The preliminary data presented in this paper cover a 12-month period, from January 1 to December 31, 2022. We believe it is useful for centres with characteristics similar to ours to report our experience and the results that can be expected in the first year of the "on-the-job training phase". If the long-term results of the ongoing training program will show relevant elements, publication of this data will follow.

How has ‘successful procedure’ been described ? Specific

  1. See 2.3. CICC Placement Procedure: Procedures were defined as successful if they resulted in correct CICC placement, regardless of the number of US-guided venipunctures performed.

Patient consent (or parental) reads somewhat bizar and contradicting between the additional info to the paper. The same holds true for the data availability statement, I cannot see the relevance of the second sentence (as also publicly available data can be use of additional research).

  1. The informed consent statement was modified. We will ask the editor for guidance on the best sentences to include and not to include.

Moreover:

  1. We have added section 2.4. Definitions and Abbreviations in order to make the manuscript easier to read.

We hope that the resubmitted manuscript responds to the reviewer's comments and suggestions.

Reviewer 2 Report

Comments and Suggestions for Authors

The authors of this manuscript analyze the partial results of a training program for neonatologists regarding the insertion of centrally inserted central catheters. In my opinion, this is solely relevant in order to point out the fact that a structured approach to any medical procedure is best. Otherwise, the intrinsic value of the manuscript is highly debatable and, as a whole, it is at least 50% longer than it should have been.

Sections

-          1.1 – the definitions (CVAD/PICC/CICC/FICC) are not necessary

-          1.2 – there are too many abbreviations, this is distracting and of no relevance for the manuscript. Also, on the second line – what is ”power injectable”? The authors should find a substitute or explain the meaning

-          1.4 – There is no need to describe every program available. It can be reduced to 4-5 lines, to include the core competencies of such programs

-          3 – I don’t find Figure 4 necessary

-          4 – the mention of the NIDCAP approach is unnecessary to the scope of the manuscript

Author Response

Thank you very much for taking the time to review this manuscript.

We also thank you for your comments and suggestions, which we have taken into account to improve the quality of the manuscript.

Please find the detailed responses below and the corresponding revisions in the re-submitted file.

-          1.1 – the definitions (CVAD/PICC/CICC/FICC) are not necessary

  1. At the beginning of the introduction (1.1., 1st paragraph), we made more explicit the terms used and their importance in the context of neonatal intensive care. For clarity of interpretation by both neonatologists and anesthesiologists, we have summarized the definitions provided by the World Congress on Vascular Access (WoCoVA) Foundation, with the intent that the reader can clearly distinguish between ECCs, PICCs, and others. We believe it is important for the reader to clearly understand the differences listed and the context of the NICUs being discussed (1.1., 2nd paragraph), because, for example, the term PICC used instead of ECC is a constant source of confusion in the scientific literature.

-          1.2 – there are too many abbreviations, this is distracting and of no relevance for the manuscript. Also, on the second line – what is ”power injectable”? The authors should find a substitute or explain the meaning

  1. We have added section 2.4. Definitions and Abbreviations in order to make the manuscript easier to read. We made explicit the meaning of power-injectable, which is used by catheter manufacturers.

-          1.4 – There is no need to describe every program available. It can be reduced to 4-5 lines, to include the core competencies of such programs

  1. In the introduction (1.4.) we listed the key WoCoVa recommendations for CVC training programs in the NICUs. The design of our training program was based on these recommendations, which are useful in explaining our methods (4.4.).

-          3 – I don’t find Figure 4 necessary

  1. Modified: the figure has been removed as suggested.

-          4 – the mention of the NIDCAP approach is unnecessary to the scope of the manuscript

  1. Modified: the mention of the NIDCAP approach was removed, but we have emphasized the “priority needs of newborns”.

Moreover:

  1. We added Supplementary Materials: Video S1, Video S2.

We hope that the resubmitted manuscript responds to the reviewer's comments and suggestions.

Round 2

Reviewer 1 Report

Comments and Suggestions for Authors

no additional comments

Author Response

Thank you again for taking the time to review our manuscript.

Reviewer 2 Report

Comments and Suggestions for Authors

Thank you for your availability to improve the manuscript. A few further points I wish to make:

In the last paragraph of section 1.1, you make the assumption that ”standard chest and abdominal radiographs (X-ray) are less accurate and less reliable than US” to assess the tip of the catheter - please provide at least one reference to support this statement. At the end of this phrase, the reference is kept as [3], although I believe it should be renumbered as [4], since you introduced a new reference.

The introduction of the new subsection with definitions and abbreviations did not have an influence on section 1.2 - it is still lengthy and difficult to read, you chose to keep the translation of all the committees and panel groups - it is a real lesson regarding the Italian language, that should only pertain to the newly introduced section 2.4

In section 2.3, the following sentence can be found (just as an example, I am positive I can find others): "The optimal catheter size was selected following the indications by the 2012 WoCoVA-GAVeCeLT-WINFOCUS consensus (WINFO-CUS stands for World Interactive Network Focused On Critical UltraSound) and the 2021 INS standards (INS stands for Infusion Nurses Society), which recommend a catheter to vein ratio of 1:3 [25]" - 1. this alternance between uppercase and lowercase letters is disconcerting and tiresome 2. the paranthesis explaining the abbreviation is unnecessary and distracting 3. the reference has nothing to do with either document.

As a whole, it took me waaay to long to read this manuscript, longer than it should have, just because of the world of distractions along the way. I sincerely hope that you can make a better job at rewriting some parts of this otherwise valuable paper.

Author Response

Thank you again for comments and suggestions. Please find the detailed responses below and the corresponding revisions in the re-submitted file.

In the last paragraph of section 1.1, you make the assumption that ”standard chest and abdominal radiographs (X-ray) are less accurate and less reliable than US” to assess the tip of the catheter - please provide at least one reference to support this statement. At the end of this phrase, the reference is kept as [3], although I believe it should be renumbered as [4], since you introduced a new reference.

  1. Thank you for correcting this typo. In reference [4] there is an extensive discussion on this topic. We changed the wording of this sentence to make it even more consistent with GAVeCeLT expert opinion [4].

The introduction of the new subsection with definitions and abbreviations did not have an influence on section 1.2 - it is still lengthy and difficult to read, you chose to keep the translation of all the committees and panel groups - it is a real lesson regarding the Italian language, that should only pertain to the newly introduced section 2.4

  1. As preferred by the reviewer, we removed references to Italian-language acronyms from the text (section 1.2.), leaving them accessible among the Definitions and Abbreviations (section 2.4.).

In section 2.3, the following sentence can be found (just as an example, I am positive I can find others): "The optimal catheter size was selected following the indications by the 2012 WoCoVA-GAVeCeLT-WINFOCUS consensus (WINFOCUS stands for World Interactive Network Focused On Critical UltraSound) and the 2021 INS standards (INS stands for Infusion Nurses Society), which recommend a catheter to vein ratio of 1:3 [25]" - 1. this alternance between uppercase and lowercase letters is disconcerting and tiresome 2. the paranthesis explaining the abbreviation is unnecessary and distracting 3. the reference has nothing to do with either document.

  1. As preferred by the reviewer: 1. we changed the standard acronyms to acronyms with only capital letters (e.g., GAVECELT for GAVeCeLT, GAVEPED for GAVePed, WOCOVA for WoCoVA, RASUVA for RaSuVa, RACEVA for RaCeVa); 2. we removed explanations of acronyms in parentheses (section 1.2.), leaving them accessible in Definitions and Abbreviations (section 2.4.).; 3. we retained the article endorsed by WINFOCUS, GAVeCeLT, and WoCoVA [25], we added reference to the latest Infusion Therapy Standards of Practice by INS [26], and we added reference to another article that better introduce the reader to the topic of the catheter-to-vein ratio [27].

We hope the reader will find the detailed introduction and methods sections useful. Moreover, we hope that the resubmitted manuscript will respond better to the reviewer's comments and suggestions.